# Freshwater budget in the Persian (Arabian) Gulf and exchanges at the Strait of Hormuz

**Edmo J. D. Campos**[1,2]*, **Arnold L. Gordon**[3], **Björn Kjerfve**[4], **Filipe Vieira**[1], **Georgenes Cavalcante**[1,5]

**1** Department of Biology, Chemistry and Environmental Sciences, College of Arts and Sciences, American University of Sharjah, Sharjah, United Arab Emirates, **2** Oceanographic Institute, University of São Paulo, SP, Brazil, **3** Lamont-Doherty Earth Observatory, Columbia University, New York, New York, United States of America, **4** School of the Earth, Ocean and the Environment, Univ. of South Carolina, Columbia, South Carolina, United States of America, **5** Institute of Atmospheric Sciences, Federal University of Alagoas, Maceió, Brazil

* ecampos@aus.edu, edmo@usp.br

**Data Availability Statement:** Campos Edmo (2020). Output from a 1/12-degree Global experiment with the Hybrid Coordinate Ocean Model (HYCOM), forced with with NCEP Reanalysis products - Data for the Persian Gulf and

## Abstract

Excess evaporation within the Persian (also referred as the Arabian) Gulf induces an inverse-estuary circulation. Surface waters are imported, via the Strait of Hormuz, while saltier waters are exported in the deeper layers. Using output of a 1/12-Degree horizontal resolution ocean general circulation model, the spatial structure and time variability of the circulation and the exchanges of volume and salt through the Strait of Hormuz are investigated in detail. The model's circulation pattern in the Gulf is found to be in good agreement with observations and other studies based on numerical models. The mean export of salty waters in the bottom layer is of $0.26 \pm 0.05\,Sv$ ($Sverdrup = 1.0 \times 10^6\,m^3\,s^{-1}$). The net freshwater import, the equivalent of the salt export divided by a reference salinity, done by the baroclinic circulation across that vertical section is decomposed in an overturning and a horizontal components, with mean values of $7.2 \pm 2.1 \times 10^{-3}\,Sv$ and $5.0 \pm 1.7 \times 10^{-3}\,Sv$ respectively. An important, novel finding of this work is that the horizontal component is confined to the deeper layers, mainly in the winter. It is also described for the first time that both components are correlated at the same level with the basin averaged evaporation minus precipitation (E-P) over the Persian Gulf. The highest correlation ($r^2 = 0.59$) of the total freshwater transport across 26˚N with E-P over the Gulf is found with a one-month time lag, with E-P leading. The time series of freshwater import does not show any significant trend in the period from 1980 to 2015. Power spectra analysis shows that most of the energy is concentrated in the seasonal cycle. Some intraseasonal variability, likely related to the Shamal wind phenomenon, and possible impacts of El-Nino are also detected. These results suggest that the overturning and the horizontal components of freshwater exchange across the Strait of Hormuz are both driven by dynamic and thermodynamic processes inside the Persian Gulf.

Strait of Hormuz. SEANOE. https://doi.org/10.17882/74042.

**Funding:** This work is funded by the American University of Sharjah Faculty Research Grant (FRG) program (Grants FRG19-M-G67 to EJDC and FRG19-M-G74 to GC). The analyses used results of numerical experiments run at the Brazilian National Institute for Space Research (INPE) Tupã supercomputer and at the Laboratory for Ocean Modeling and Observations (LABMON), of the Oceanographic Institute of the University of São Paulo, Brazil, as part of Projects SAMOC-BR and SAMBAR, sponsored by the São Paulo State Foundation for Research Support (grants 2011/50552-4 and 2017/09659-6) to EJDC. EJDC acknowledges the Brazilian Council for Scientific and Technological Development (CNPq) for a Research Fellowship (Grant 302018/2014-0).

**Competing interests:** The authors have declared that no competing interests exist.

# 1 Introduction

The Persian (or Arabian) Gulf, hereinafter referred as "the Persian Gulf" or simply "the Gulf", and the Red Sea inject into the Indian Ocean waters with the highest salinity of the world oceans [1–3]. The volume of water injected by those semi-enclosed seas is small: only 0.2 Sv for the Persian Gulf water and 0.4 to 0.8 Sv for the Red Sea water [3, 4]. Nevertheless, due to their unique thermohaline characteristics, waters from these two marginal seas can be tracked from their sources over great distances, spanning a large area of the northern Indian Ocean [2, 4–9]. In spite of the high salt content, waters from the Gulf and the Red Sea are warmer and more buoyant than the Indian Ocean Deep Water, which make them incapable to reach depths greater than 1000m. However, by mixing with the thermocline waters and the deeper waters below, the waters from the two marginal seas can modify the water mass properties in a large area of the northern and equatorial Indian Ocean.

Despite their small volumes, due to their high salinity content and the ubiquitous presence in a large area of the Indian Ocean at thermocline levels, the waters from the Red Sea and the Persian Gulf may have significant influence in the modulation of the Indian Ocean water masses [2, 6, 8]. However, to what extent these water really impact the larger scales is still an open question and is the subject of ongoing research at the American University of Sharjah. Preliminary results indicate that, indeed, the Persian Gulf and the Red sea waters play relevant roles in dynamic and thermodynamic processes in the Arabian Sea and in the Bay of Bengal. In any case, additional knowledge of their main general patterns and spreading in the Indian Ocean is crucial to better understand the seasonal and longer-scales variability of the water flow at the entry points into the Arabian Sea. In this particular, one key question relates to the impacts that increased evaporation due to global warming [10–13] and the human induced changes including the intense activity of water desalination [14–17] in the Persian Gulf, would have on the Arabian Sea and Indian Ocean [10, 11]. Of course, these changing processes are common to the Red Sea as well [18]. However, the Red Sea has a much deeper basin, connected to the Gulf of Aden by the Bab el Mandeb, a relatively shallow and narrow channel. Waters with increased salinity tend to sink to greater depths and be trapped inside the basin, with a mean deep-water residence time of approximately 36 years [18]. On the contrary, the Persian Gulf is much shallower, with depths increasing towards the Strait of Hormuz and the Sea (or Gulf) of Oman. Changes within the Persian Gulf are communicated directly to the Indian Ocean on much shorter time scales. In his classical paper, *Reynolds (1993)* [1] comments that estimates of the mean residence time of the Gulf, or the time a water parcel would take to leave the basin, vary on an approximate range of 2 to 5 years. More recent studies report shorter residence times. *Xue and Eltahir (2015)* [19], based on results of a coupled ocean-atmosphere model, found a mean residence time of approximatelly 14 months. *Alosairi et al. (2011)* [20] used a numerical model based Lagrangian approach to compute the residence time of passive tracers released in different locations in the Gulf. The results show a wide spatial variation, with times ranging from approximately 1000 days near Kuwait to only 2 days at the vicinity of the Strait of Hormuz.

Recent studies suggest that the Persian Gulf might be going through changes. Based on descriptive analyses of daily optimum interpolated sea surface temperature anomaly distributed by the USA National Oceanic and Atmospheric Administration (NOAA), *Noori et al. (2019)* [11] report a gradually increasing trend in sea surface temperature (SST) in the Gulf and the Gulf of Oman since the 1980's. In another investigation, using an autoregressive integrated moving average model, *Shirvani et al. (2015)* [21] found that, as a result of global warming, the averaged SST in the Gulf has increased by 0.57˚C from 1950 to 2010, with a more pronounced trend in the last two decades. Considering the overall importance of the

Gulf water, these changes in the Gulf may have significant consequences to the Indian Ocean circulation.

Considering the ubiquitous presence of the Persian Gulf waters in the upper-thermocline layers of the northern Indian Ocean, it is natural to question what would be the consequences of possible changes in the Gulf. To answer that question, it is crucially important to know better the present-day climatological conditions and time-variability of the Gulf's general circulation, the salt export through the Strait of Hormuz and the relationship with dynamic processes and the freshwater budget in the Persian Gulf. This motivated the present study, which uses results of a numerical experiment with the Hybrid Coordinate Ocean Model (HYCOM) to investigate the spatial structure, the seasonality and interannual variability of the freshwater budget in the Persian Gulf, and the exchanges of water and salt with the Gulf of Oman.

## 2 Circulation in the Persian Gulf

Aspects of the climatological conditions and seasonality of the Persian Gulf circulation have been described by modeling and observational studies in the past few decades [1, 3, 19, 22–28]. According to the literature, the Persian Gulf is a semi-enclosed marginal sea with mean depth of 36m, connected to the Indian Ocean through the narrow Strait of Hormuz and the Gulf of Oman. It is located in a region with arid, subtropical climate, with evaporation much greater than freshwater inflow. As a result of the net freshwater loss, it presents a reverse estuarine circulation, importing Indian Ocean Surface Waters (IOSW), in the top layers, and exporting higher salinity waters, in the bottom layers, through the Strait of Hormuz. These density currents are central in the dispersion and removal of salt and other suspended material from the Gulf. Tidal forcing impacts circulation to a minor extent and only on the smaller scales of space and time [27–29].

Wind forcing is significant, particularly the Shamal weather phenomenon [1, 3, 23]. Shamal is a northwesterly wind that blows over the Gulf mostly in summer but sometimes in winter as well. Summer Shamal winds are important in driving the mid-Gulf eddies and transporting water to the southern basin [30]. They have ecological importance in transporting larvae as well as evaporative cooling of the extremely hot summer sea temperatures [30, 31].

In spite of the details added in later studies, the general features of the mean circulation in the AG follow the schematics presented by *Reynolds (1993)* [1]. According to that general representation, and the literature indicated above, the mean motion is dominated by a barotropic cyclonic gyre. Near the surface, the less saline IOSW enters the Gulf through the Strait of Hormuz forming two branches. One, to the north, flows along the Iranian coast and in summer can reach regions to the west of Qatar, forming there a cyclonic circulation. The return flow is by mean of a coastal current along the Arabian and the United Arab Emirates coastlines. The other branch of the waters entering through the Strait of Hormuz turns southward is driven mainly by the Ekman drift due to the predominant northwesterly winds [3, 23]. It crosses the basin and reaches the southern coasts of the Arabian Peninsula, forming an overall cyclonic circulation in the southern portion of the Gulf. This general circulation pattern persists throughout the year with marked seasonal variation. In wintertime, opposed by stronger northwesterly winds, the northern-branch inflow does not penetrate past 27˚N. As the winds relax in the spring and summer, and the seasonal thermocline is established, the current is coastally intensified and can reach the northern end of the Gulf, bringing the tongue of low-salinity water up to 28˚N. The southern branch is strongest during the winter. It is driven by the combined effect of the stronger winds and by the sinking of the dense water in the southern Gulf. In its spreading into the Gulf, the IOSW waters entering through the two branches undergo a considerable salinity increase by evaporation and mixing with ambient saltier water.

## 2.1 Freshwater budget and exchanges with the Gulf of Oman

The present knowledge on the freshwater budget in the Gulf and the exchanges of water and salt between the Persian Gulf and the Gulf of Oman is based mainly on results of numerical models, with much fewer observational studies. Regarding to the latter, one of the most comprehensive observational effort was the expedition on board the US NOAA RV Mt. Mitchell in the region comprised by the Persian Gulf, the Strait of Hormuz and the Gulf of Oman from February to June 1992 [1]. Other relevant observational studies are [3, 26, 32].

The Mt Mitchell Expedition provided the data used by the *Reynolds (1993)* [1] seminal work, which produced an integrated picture of the Gulf's physical oceanography. Based on the data collected during the Mt Mitchell Expedition, ancillary data and literature review, *Reynolds (1993)* [1] produced an up-to-date accurate knowledge on the Gulf freshwater budget. Evaporation is much greater than the precipitation. Estimates of evaporation were in the range from 144 $cmyr^{-1}$ to 500 $cmyr^{-1}$, with most of the evaporation occurring in winter, Precipitation is of the order of 7 $cmyr^{-1}$. According to *Reynolds (1993)* [1], the higher winter evaporation rates are due to the high wind speeds associated with the strong winter Shamal.

Annual averages of the major rivers outflow were estimated as follows: The Shat al-Arab, comprising the Tigris, Euphrates and Karun, 1456 $m^3 \, s^{-1}$; the Hendijan, 203 $m^3 \, s^{-1}$; the Hilleh, 444 $m^3 \, s^{-1}$; and the Mand, 1387 $m^3 \, s^{-1}$. If distributed over the entire Gulf Area (251,000 $km^2$), the combined outflow of these rivers would equate to a precipitation of approximately 44 $cmyr^{-1}$. In his paper, however, Reynolds warns that one should be cautious and further verify these river runoff estimates, which were based on Iranian river measurement reports and could be overestimated. Reynolds [1] also points out the complexity of the circulation at the Strait of Hormuz and very little is described of the salt exchanges with the Gulf of Oman.

Based on observational work [1, 3, 26, 32], the circulation at the Strait of Hormuz has a two-layer structure, with the inflow of fresher IOSW in the upper layer and the outflow of saltier water along the bottom. Data from a current meter mooring deployed at 26° 16'N, 56° 05'E show that, while in the upper layer the flow has a marked seasonal variability, the bottom flow is relatively steady with speed in the range of 0.2 to 0.3 $ms^{-1}$ [3]. In another observational work [26], data from a moored ADCP at 25° 36.1'N, 57° 00'E covering the depth range from 23$m$ to 115$m$ show a highly variable flow, with along-isobath and cross-isobath speeds in the range of –0.35 to 0.35 $ms^{-1}$ and –0.55 to 0.45 $ms^{-1}$, respectively. Both articles, [26] and [3], report geostrophic volume transports in the inner part of the Strait of Hormuz in the order of 0.22 $Sv$ to 0.24 $Sv$.

According to the schematic representation in *Johns et al. (2003)* [3], the upper layer inflow supplies water for two branches of the circulation inside the Gulf. One part will flow to the northernmost regions, where it sinks and form the outward bottom flow. The other recirculates in a horizontal gyre-like circulation. Based on that, the authors [3] suggest that the mean exchange through the Strait of Hormuz is composed by a horizontal flow superimposed on the vertical inverse estuarine circulation. However, *Johns et al. (2003)* [3] point out that their observations, restricted to the inner (or western) part of the Strait, do not allow inference whether this horizontal flow actually reaches the Gulf of Oman or just recirculates within the Persian Gulf. More recently, observational data collected by *Ghazi et al. (2017)* [32] on a zonal section on the eastern part of the Strait (26°N), show that the bottom outflow is confined to the western side while the upper inflow is dislocated towards the east.

The knowledge based on the scarce observations has been confirmed and extended by studies based on numerical models. Results of one of the most relevant of these numerical efforts, motivated by observational work of *Johns et al. (2003)* [3], were published in two scientific papers: *Yao and Johns (2010a)* [23] and *Yao and Johns (2010b)* [24]. Using an implementation

of HYCOM, *Yao and Johns* [23] [24] run a series of experiments to study the seasonal variability and the sensitivity to different products. Forcing the model with the climatological COADS (Comprehensive Ocean-Atmosphere Data Set) products, they found that the simulated volume transport in the bottom layer, contrary to the fairly steady values sampled by *Johns et al. (2003)* [3], presented significant seasonality, being larger in the summer than in fall and winter. The maximum transport occurred in July, with value of 0.15 *Sv* and the minimum of 0.07 *Sv* in January. In *Yao and Johns (2010a)* [23], the estimate for the annual mean of volume transport of the bottom flow is of 0.12 *Sv*, weaker than the 0.15 *Sv* reported by *Johns et al. (2003)* [3]. In experiments forced with buoyancy-only forcing and in runs forced with high-frequency atmospheric products they found similar seasonal cycle as in the COADS run. The study by Yao and Johns [23] allows the conclusion that forcing the model with high-frequency atmospheric products results in a more realistic surface temperature distribution but does not change significantly the general features of the circulation.

## 3 Materials and methods

### 3.1 The numerical model

The study is based on the analysis of results of a numerical experiment run with HYCOM [33, 34], the same model used by [23, 24]. It is a widely known model and here a brief description is given, just to highlight the usefulness of it's layered coordinate system to the present investigation.

HYCOM is a layered ocean model wherein the ocean is represented as a stack of shallow water layers (Fig 1). Unlike traditional ocean models the vertical coordinate is not depth (or pressure) but is a combination of depth (pressure), sigma (bathymetry-following) and/or isoentropic (isopycnal) surfaces, which varies with time [33, 34]. As it will be pointed out in the manuscript, this layered system turns out to be very convenient to study processes in regions such as the Persian Gulf and the Gulf of Oman.

For this study, the model products for the Gulf area were extracted from results of a numerical simulation, hereinafter referred as GLBa0.18, executed at the Ocean Numerical Modeling Laboratory (LABMON) of the University of São Paulo (Brazil). In GLBa0.08 the model was run in a global domain discretized on a 2-D horizontal PANAM mesh: a combination of a

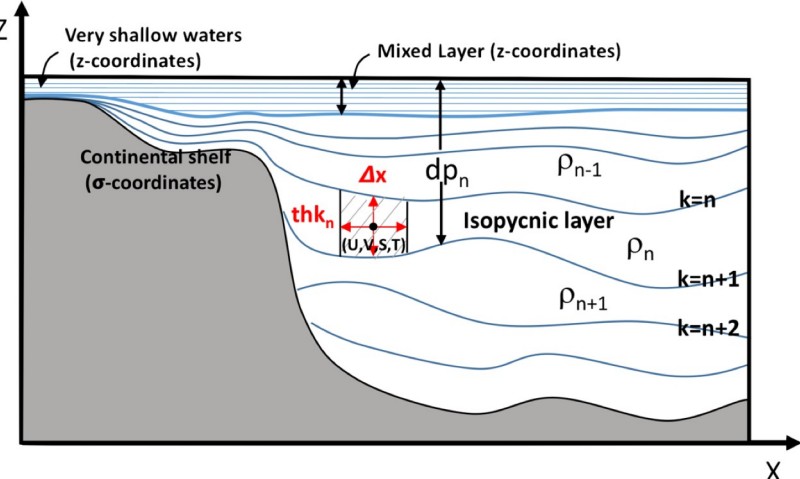

**Fig 1. Schematic representation of the model's hybrid vertical coordinate system.** $dp_n$ is the depth of the $n_{th}$ interface. In ocen interior, they are the bottom of the layer $k = n$, with density $\rho_n$.

Mercator grid south of 47˚N with a double patch to the north of that latitude. In horizontal surfaces, the grid spacing is equal to 1/12 of a degree at the equator but varies with the latitude, resulting in grid-sizes of about 7$km$, on average (detailed information on HYCOM global implementations is available at https://www.hycom.org/global). The grid's vertical structure is formed by 32 hybrid layers, allowed to change from isopycnic to "z" or terrain-following surfaces as the model runs. The bathymetry was based on the ETOPO-5 data set [35]. The model was forced with time-varying monthly means of the NCEP-1 Reanalysis products [36] from 1949 to 2015. Tidal forcing was not considered—even though tides in the Gulf are important in the short term, they do not generate significant residual currents in longer time scales [27, 28] in the Persian Gulf.

For generating initial conditions, the following strategy was adopted. First, a "warm-up" experiment was run, using as initial condition the output from an experiment downloaded from the HYCOM Consortium homepage, for an arbitrary day. That is to say, the model was not started from rest but, rather, with all variables already in some level of dynamical equilibrium. Then, the model was run for 27 years, forced with climatological products from NCEP. The output for the last day of this "warm-up" run, corresponding to day 360 of the 27th year, was then used to initialize the dated run, starting on 1/Jan/1949. The dated run extended from from Jan/1949 to Dec/2015. Six-days averages of the model's output for the last 35 years (1980 to 2015) were considered in the present analysis.

## 3.2 The atmospheric and radiative forcing

The following variables from the NCEP-1 Reanalysis were considered for forcing the model: surface air temperature; the surface net downward radiation flux; surface net downward short-wave flux; precipitation; vapor mixing ratio; surface wind speed and wind stress. The mean values, standard-deviations and trends for the NCEP forcing fields, in the period 1980-2013, are shown in Table 1. Most of the variability for all forcing components is in the seasonal cycle. In particular, the basin-averaged time series of net short-wave radiation, air temperature and wind speed over the sea surface have a pronounced variation within a climatological year, with the higher values during the summer months. The maximum values of the short-wave radiation and wind speed occur around the same time, in June, while the air temperature peaks one month later, in July (Fig 2).

## 3.3 Approach for the computation of lateral freshwater (or salt) exchange

As described in the Introduction, due to the much greater evaporation than freshwater input by precipitation and river inflow, the Persian Gulf is a hypersaline environment. The net evaporation drives salt export or, equivalently, freshwater import from the Indian Ocean through the Strait of Hormuz (SH). As suggested in the results of [3] and [23], in the inner part of Gulf, this lateral exchange of freshwater (or, equivalently, salt) done by the large-scale, tri-dimensional flow can be divided in two components: a vertical overturning and a horizontal

**Table 1. NCEP forcing fields statistics for the period 1980 to 2013.**

| Field | Mean | St. Dev. | Trend ($year^{-1}$) | MK test (95%) |
|---|---|---|---|---|
| Wind Speed | 2.8 $ms^{-1}$ | 0.6 $ms^{-1}$ | $-0.01\pm.01ms^{-1}$ | 1 |
| Air Temp. | 25.6˚C | 7.3˚C | $0.03\pm0.16$˚C | 0 |
| Rad. Flux | 219.5$Wm^{-2}$ | 57.2$Wm^{-2}$ | 0.00 | 0 |
| Sh. Wave Flux | 104.1$Wm^{-2}$ | 41.5$Wm^{-2}$ | $-0.04\pm0.89Wm^{-2}$ | 0 |
| Precipitation | 0.10 $myear-1$ | 0.13$myear^{-1}$ | 0.00 | 0 |

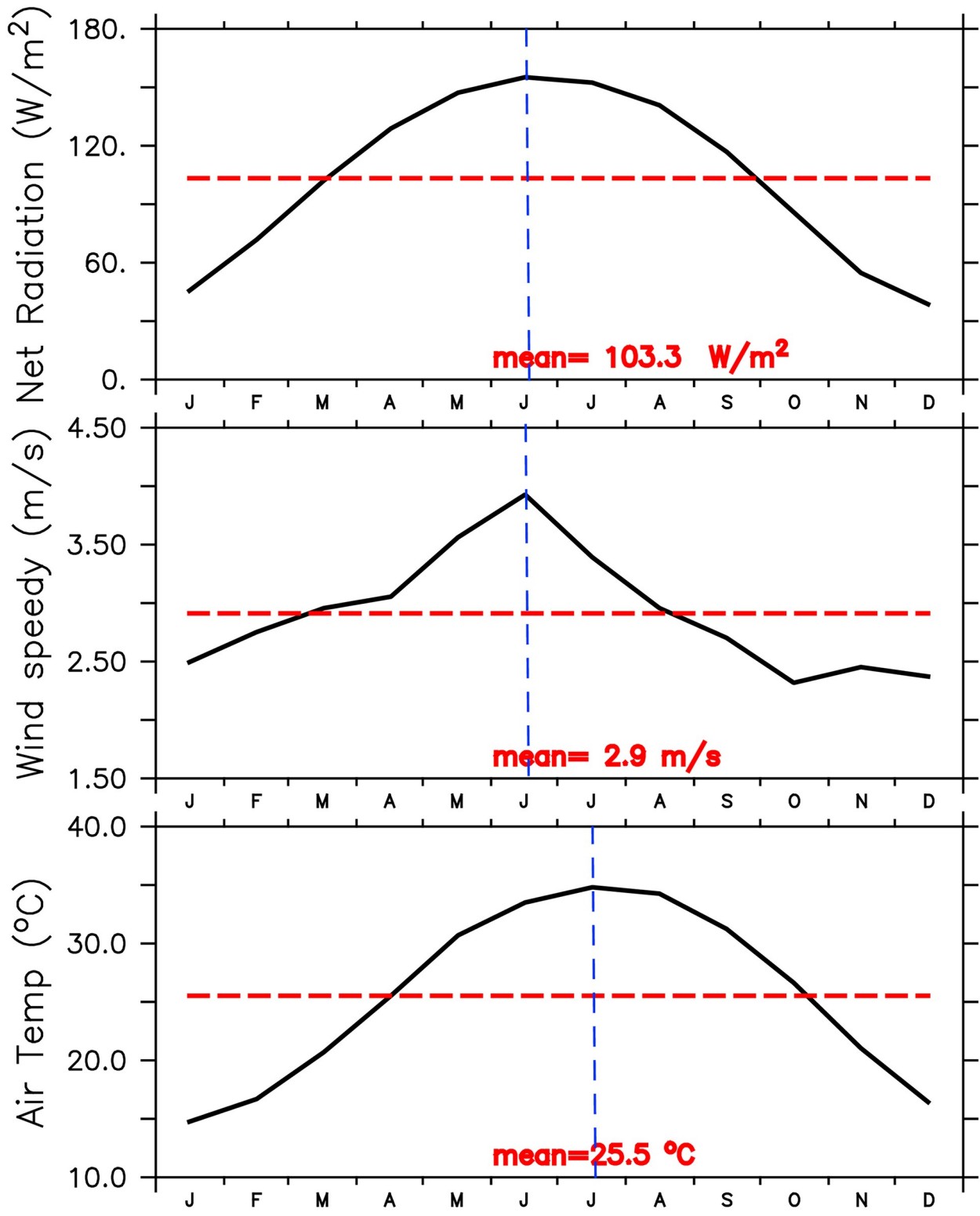

**Fig 2. Seasonal variation of the basin-averaged net shortwave radiation, wind speed and air temperature form NCEP.**

circulation. In a numerical simulation, a third component is related to sub-grid processes not resolved by the model. The better understanding of this mechanism is highly relevant since any imbalance could lead to salinity trends in the Gulf with possible impacts on the hydrographic and environmental properties, both locally and in a vast area of the northern Indian Ocean.

Here, the tri-dimensional characteristic suggested by *Johns et al. (2003)* [3] is investigated, considering that the Persian Gulf can be regarded as a small-scale analogue of the Atlantic Ocean basin. Equivalently to the input of waters from the Bering Strait to to North Atlantic, through the Arctic, in the Gulf there is the riverine inflow of waters from the Shat al-Arab. A zonal section along 26˚N, between Oman and Iran in the outside part of the Strait of Hormuz, can be associated to the 34.5˚S line extending from South America to Africa. Similar to what happens across 34.5˚S in the Atlantic, the baroclinic circulation across the Strait of Hormuz can be considered as a sum of an overturning and a gyre-like horizontal component described by *Johns et al. (2003)* [3]. For this reason, and also because the same reasoning could be applied to larger regions of the Indian Ocean, here the following approach, adapted from *Drijfhout et al. (2011)* [37], was used to compute the total volume and salt (or, equivalently, freshwater) budgets for the Persian Gulf.

Start considering $v_H(x, z, t)$ as the meridional component of the velocity across and $S_H(x, z, t)$ as the vertical distribution of salinity at the Hormuz section. Then, the Conservation Laws for the total volume (V) and the total amount of salt (S) inside the Persian Gulf can be expressed as:

$$\mathbf{V_t} = \int \int v_H dx dz + EMP + Rivers + R_V \tag{1}$$

and

$$\int \int \int S_t dV + \mathbf{V_t}\bar{S} = \int \int v_H S_H dx dz + Source + Mix + R_S. \tag{2}$$

In these equations,

$$\mathbf{V_t} = \frac{\partial V}{\partial t}$$

is the total volume trend;

$$S_t = \frac{\partial S}{\partial t}$$

is the total salinity trend;

$$\bar{S} = \frac{1}{V} \int \int \int S dV$$

is the volume-averaged salinity and *EMP* is the evaporation minus precipitation over the entire area of the Persian Gulf. *Source* represents the net effect of sources and sinks of salt; *Mix* is the effect of small scale mixing at Hormuz and *RV* and *RS* are residual terms due to truncation. Note that the volume trend $\mathbf{V_t}$ was written in bold face just to highlight the link between the two equations.

Next, define the quantities:

$$M_t = \frac{-1}{S_0} \int \int \int S_t \, dx\,dy\,dz \tag{3}$$

$$M_{ov} = \frac{-1}{S_0} \overline{(v^*)} < S > dz \tag{4}$$

$$M_{az} = \frac{-1}{S_0} \int \overline{v'S'} \, dz \tag{5}$$

where:

$$\tilde{v} = \frac{\int v_H \, dx\,dz}{\int dx\,dz} \rightarrow \text{Barotropic Velocity}, \tag{6}$$

$$v^* = < v_H > - \tilde{v} \rightarrow \text{Baroclinic Velocity}, \tag{7}$$

$$S_0 = \int \int S_H \, dx\,dz \Big/ \int \int dx\,dz \tag{8}$$

and

$$< v_H > = \frac{\int v_H \, dx}{\int \int dx}, v' = v_H - < v_H > \tag{9}$$

In Eqs (4) and (5), the overline denotes zonal integration; $M_t$ is the total freshwater-equivalent tendency inside the Gulf; $S0$ is area averaged salinity at Hormuz; $M_{ov}$ and $M_{az}$ are, respectively, the freshwater transports associated with the overturning and azimuthal (horizontal) components of the baroclinic circulation across the Strait of Hormuz.

## 4 Results and discussions

### 4.1 The Gulf circulation

The GLBa0.08 experiment was designed to study processes related to the large-scale circulation. Nevertheless, because of its relatively high horizontal resolution (1/12-degree) and the use of a satisfactorily realistic bathymetry [35], it is reasonable to expect that the results could be used to study some aspects of the circulation in a smaller region such as the Persian Gulf. To check this assumption, a comparison of the circulation pattern produced by the model in the Gulf was made with results of previous studies, including numerical simulations carried out with the same model (HYCOM) [1, 23–25, 28]. For better compatibility, seasonal averages were computed for the same periods as in the cited publications: January-February-March (JFM), April-May-June (AMJ), July-August-Septermber (JAS), October-November-December (OND).

It is found that the model results agree reasonably well with general aspects of the Gulf's circulation described in the literature. The analysis of GLBa0.08 output shows the mean circulation pattern (Figs 3 and 4) is similar to that described in the classic work of *Reynolds (1993)* [1] and the other studies cited above. The circulation is dominated by a year-round barotropic cyclonic gyre with a significant first baroclinic mode. In the top layer, represented by the vertical average in the upper seven layers of the model (mean depth of approximately 25m), the less

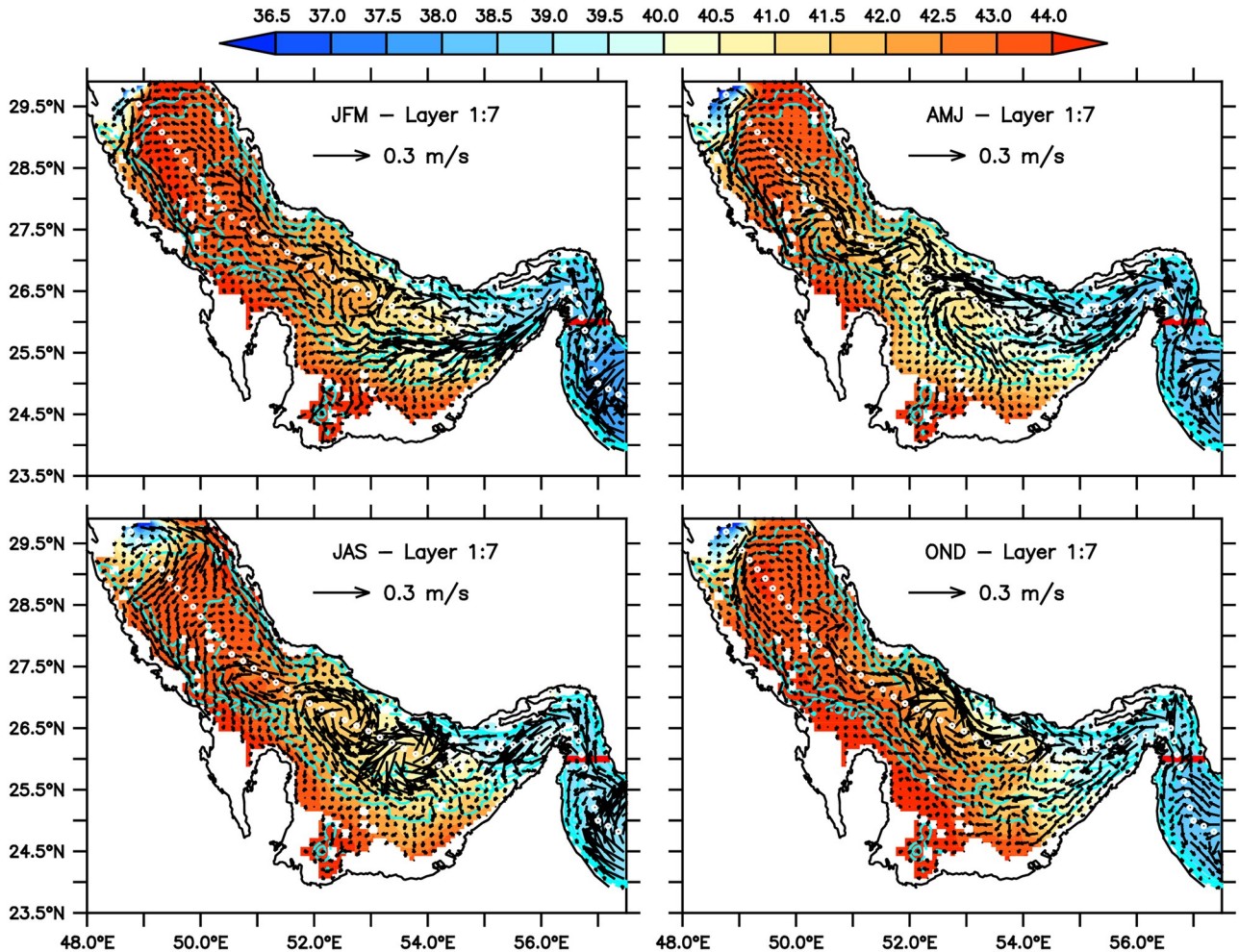

**Fig 3. Seasonal mean distributions of salinity and velocity averaged over the upper seven layers of the model (mean depth of 25m).**

saline waters from the Gulf of Oman enter the Persian Gulf through the Strait of Hormuz forming two branches, which behaves quite similarly to the pattern described in literature (Fig 3). The northern branch flows along the Iranian coast, reaching the northwestern regions of the Gulf. The southern branch flows towards the coasts of the Arab Emirates and seems to be most affected by the seasonality, being stronger in the summer. This finding is in agreement with *Cavalcante et al. (2016)* [30] and *Thoppil et al. (2010))* [25]. In the winter season (JFM), the top layer circulation, shown in the upper left panel of Fig 3, has the two branches well defined and with weak mesoscale activity. Starting in the following season, AMJ, strong meso-scale eddies start to form and remain for the other two seasons, JAS and OND. A strong quasi-stationary cyclonic vortex appears around 51°E—53°E in AMJ, peaks in the summer (JAS) and starts to decay in the autumn (OND). These mesoscale features are also present in the cir-culation patterns obtained by [23].

The seasonal circulation and salinity distribution for the four seasons in the bottom layer, the vertical average over the model's layers 8 to 13, are shown in Fig 4. In spite of some signa-ture of the barotropic features, such as the mid-basin cyclonic vortex in JAS, it is quite clear the reverse estuary characteristic of the circulation. The overall salinity is higher, and the pre-dominant direction of the flow is towards the Strait of Hormuz.

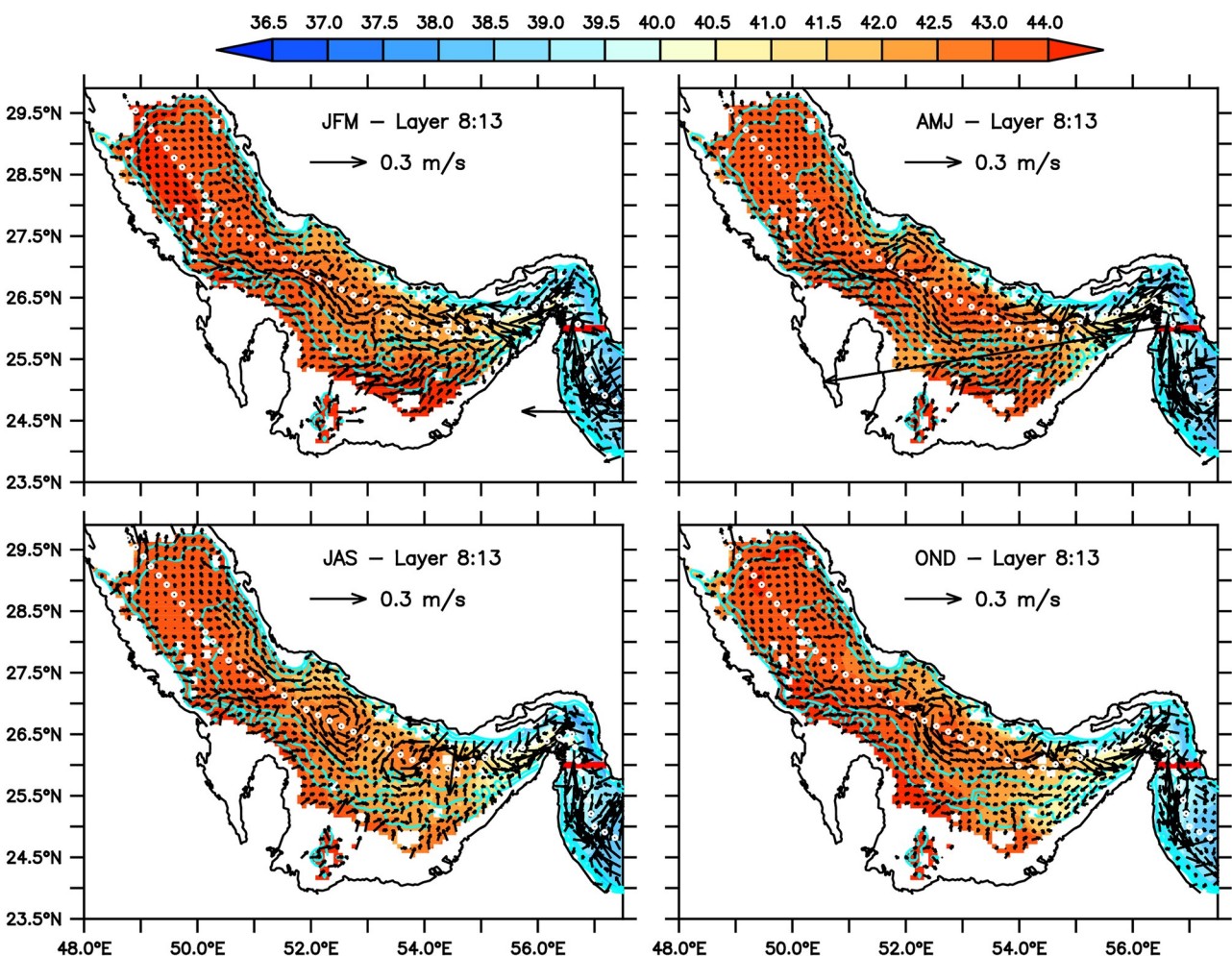

**Fig 4. Seasonal mean distributions of salinity and velocity averaged from the 8th to 13th seven layers representing the circulation and salinity in the "bottom layer", with a mean thickness of about 20m (see Fig 5).**

The two-layer, inverse-estuary circulation in the Gulf is well represented by the vertical structure of the seasonal temperature and salinity along an axial section starting near the Shat al-Arab and following the deeper channel towards the Gulf of Oman, indicated by the sequence of small white circles in Figs 3 and 4. As shown in Fig 5, the isopycnic surface $\sigma_2 = 32.60 kgm^{-3}$, the bottom of the 7th layer of the model, clearly separates the two regions. Fig 5 also shows that, from within the Persian Gulf basin, passing through the Gulf of Oman and in the Arabian Sea (beyond the longitude 60˚E), waters with the model's Gulf water characteristics ($18.5˚ C < T < 27.9˚ C$ and $S > 37.7$) is mainly confined between the 7th and the 13th isopycnic surfaces ($\sigma_2 = 32.60$ and $35.15 kgm^{-3}$). This once more shows how suitable a layered coordinate model, such as HYCOM, is for studying this circulation pattern, minimizing the effects of spurious numerical diffusion, for instance.

## 4.2 Comparing the results with observations in the Strait of Hormuz

One of the main purposes of this investigation is a better understanding of the spatial structure and time variability of the exchanges between the Persian Gulf and the Gulf of Oman, using the GLBa0.08 numerical experiment. As such, in order to evaluate the model's

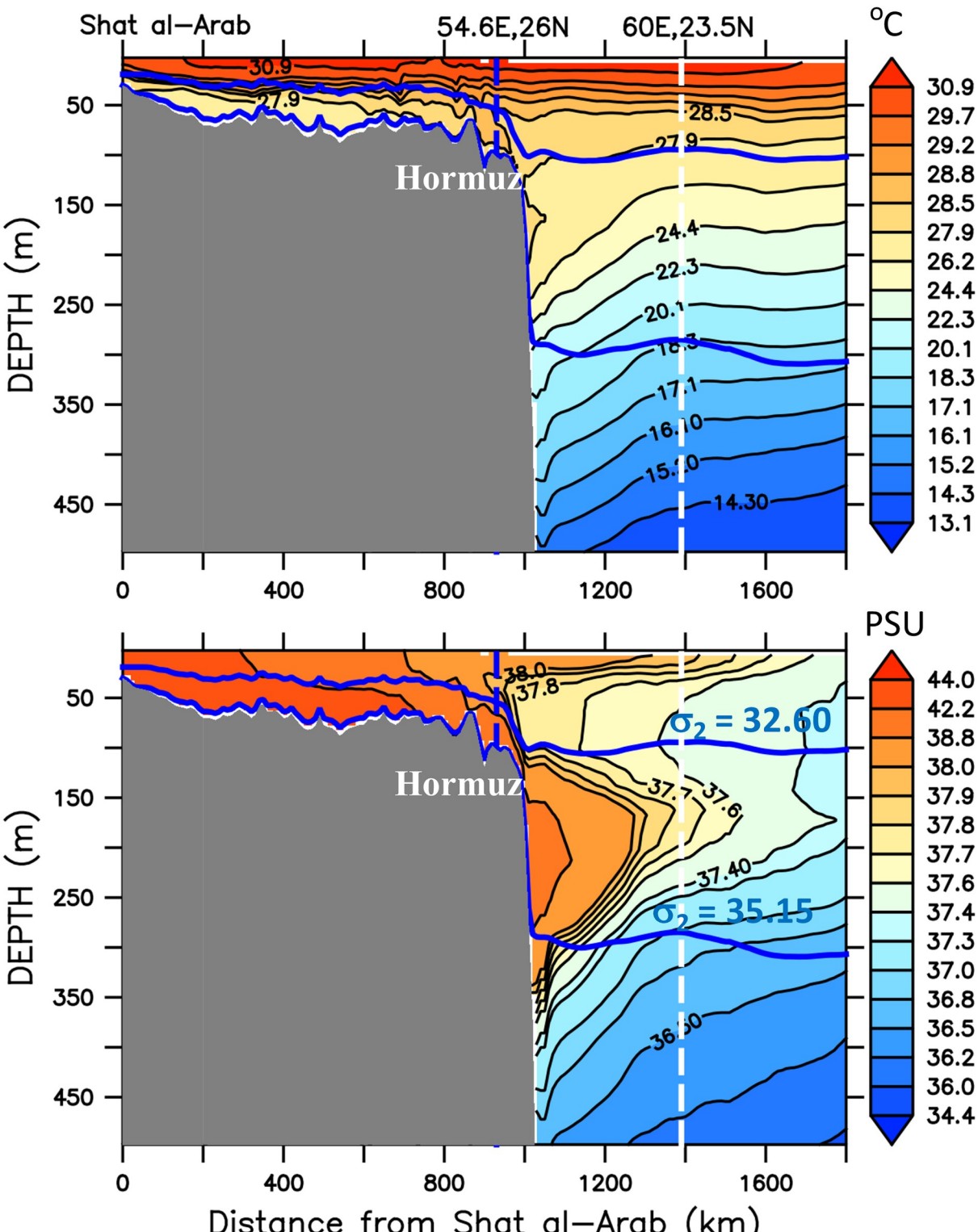

**Fig 5. Mean temperature and salinity distributions on a vertical section along the line defined by the sequence of small white circles in Figs 3 and 4.** The two blue lines represent the isopycnic surfaces $\sigma_2 = 32.60$ and $\sigma_2 = 35.15$, which defines the thermocline in the deep ocean. The upper one represents the interface between the upper and bottom layer inside the Gulf.

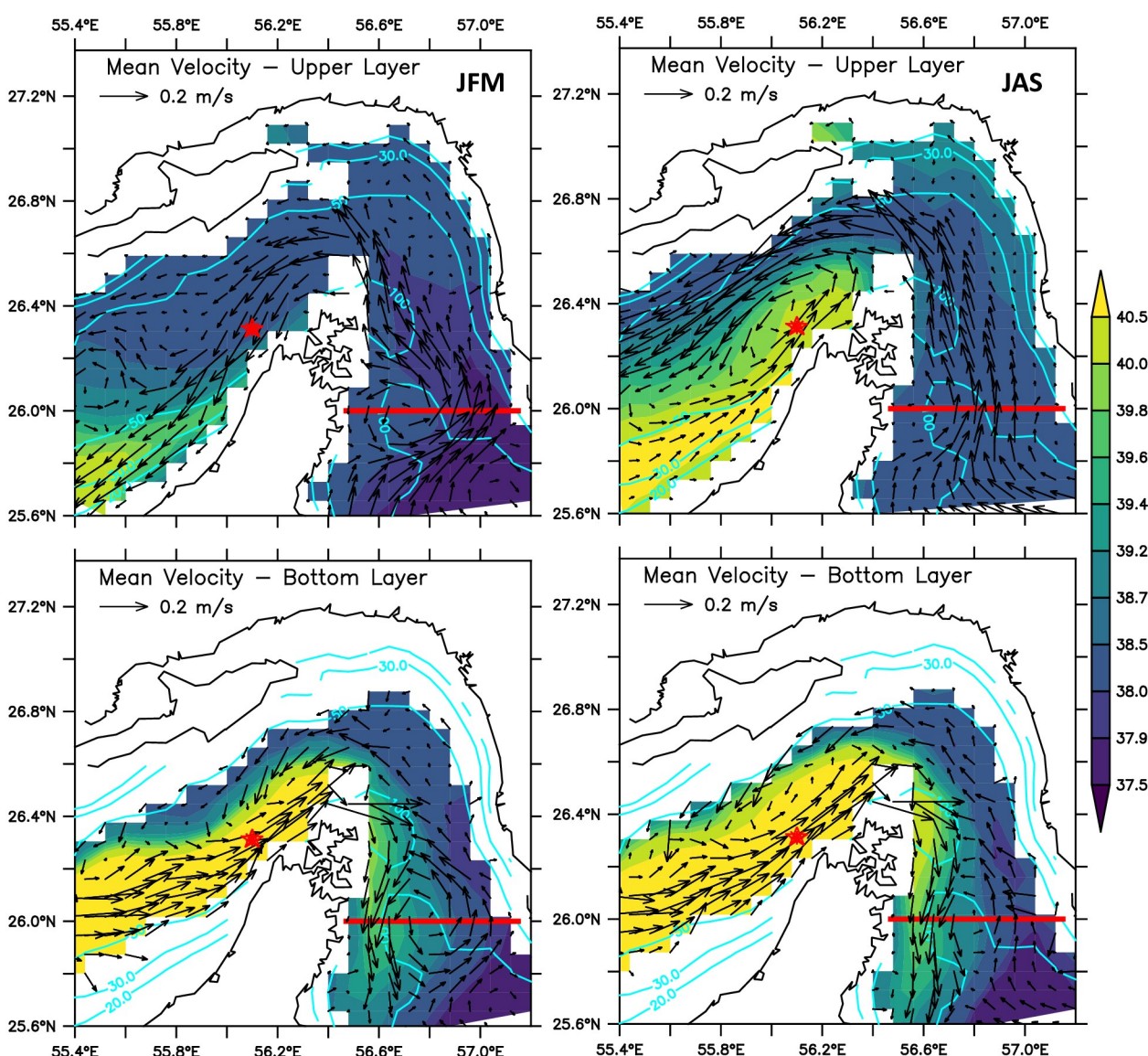

**Fig 6. Salinity and velocity distributions for Winter (JFM) and Summer (JAS), in the upper (top panels) and lower layers (bottom panels), in the region encopassing the Strait of Hormuz.** The red star indicates the [3] mooring site. The thick red line along 26°N, from 56°50'E to 57°10'E indicates the zonal section where some diagnostic are coomputed in the following Sections.

accuracy, a more detailed look at the two-layered circulation in the Strait of Hormuz is made, to compare with results of other numerical experiments [23, 24] and with observed data sampled by instruments moored at 56.08°E, 26.27°N [3]. Fig 6 shows the mean salinity map and the velocity vectors at each of the model's grid-points, for the winter (JFM) and summer (JAS) seasons, in the top (0-40m) and bottom (50-90m) layers. The pale-blue lines are isobaths and the red star indicates the [3] mooring site. The thick red line along 26°N, from 56°50'E to 57°10'E indicates the zonal section where some diagnostic will be computed in the following Sections.

According to Fig 6, during the winter time (JFM) the circulation produced by GLBa0.08 in the Strait of Hormuz has a well defined reverse estuary circulation, with fresher water entering

the Persian Gulf in the top layer and saltier flowing towards the Gulf of Oman in the lower layer. In the summer (JAS) a well defined recirculation cell, appears near the western region of the Strait of Hormuz. This pattern is not entirely in agreement with Johns et al. (2003) [3] observations, which showed a persistent recirculation cell throughout the year. In the GLBa0.08 results, the recirculation seems to be restricted to the inner part of the Strait of Hormuz, with no signal of upper layers waters leaving the Persian Gulf towards the Gulf of Oman. On the other hand, as shown in the lower two panels of Fig 6, the circulation in the bottom layer is composed by an intense outflow along the Arab Emirates and Oman coastline and a less intense inflow in the northern part, along the Iranian coast. This clearly supports the idea that the exchange beetween the Gulf and the Gulf of Oman is composed by overturning and a horizontal components.

To better compare the vertical and temporal structures of the GLBa0.08 flow with the data collected by [3], stickplots of velocity were plotted at depths 10m, 20m, 40m, 60m and 80m for the period 1/Dec/1996 to 1/Jan/1999 at the location of the mooring site of [3] (Fig 7). According to the model, the upper 20m meters are dominated by a flow predominantly from the NE, with short inversions peaking during the JAS season, as seen in Fig 6. At 40 m, despite the seasonal variability, the currents start to show a tendency to flow more towards the NE. These results show that structure of the currents produced by the model in the upper layer is not entirely in agreement with the time series measured at the mooring site. The observations show that, in spite of intense high-frequency variability, the overall direction of the flow in the upper layers is to the N-NE, in all seasons. A possible explanation for the differences is that the model is not reproducing correctly the recirculation branch of the upper flow, which according to the observations, occur in the vicinity of the Arabian coastline (where the mooring was located). This is likely to be an effect of the model's coarser bathymetry in the region. Also, it should be noted that only 6-days averages of the model products were available. Thus, one should not expect any good agreement with the higher frequencies present in the data. In the bottom-layer, however, there is a much better agreement with the observations. Below 40m, the flow is relatively steady, as measured by [3]. The stickplots of Fig 7 show that the flow is relatively constant and oriented exclusively toward the NE, with average speeds in the order of 20-25cm/s, comparable with the observations.

The results of GLBa0.08 were compared with the numerical experiments run by [23]. In the latter, the upper-layer flow forced with climatological products also do not show a significant recirculation in the region of the mooring. As much as in GLBa0.08, in [23] the inflow in the top layer is located closer the coast and is predominantly to the SW. In the run forced with high-frequency atmospheric products, the more intense upper-layer flow is dislocated towards the Iranian coasts and some recirculation appears near the Arabian Peninsula. In GLBa0.08 the bottom flow also shows sign of seasonal variability, in agreement with the results of [23]. Another interesting feature seen in Fig 6 is the presence of a cyclonic circulation cell, more evident in the deeper layers, both in JFM and JAS.

Due to the relatively coarse horizontal resolution and quality of the model's bathymetry, a decision was made to locate a section for the diagnostics of the lateral exchanges at the latitude 26˚N, in the outer region of the Strait of Hormuz. In the model's results, the upper-layer recirculation described in [3] does not extend beyond the Musandam peninsula, towards the Gulf of Oman. However, as seen in Fig 6, the bottom flux is at the entrance of the Strait is formed by inward and outward flows. This reforced the relevance to investigate the tri-dimensional structure of the flow in the outer part of the Strait of Hormuz, considering the presence of a horizontal component of the circulation. The zonal section along 26˚N hereinafter will be sometimes referred as the Hormuz section.

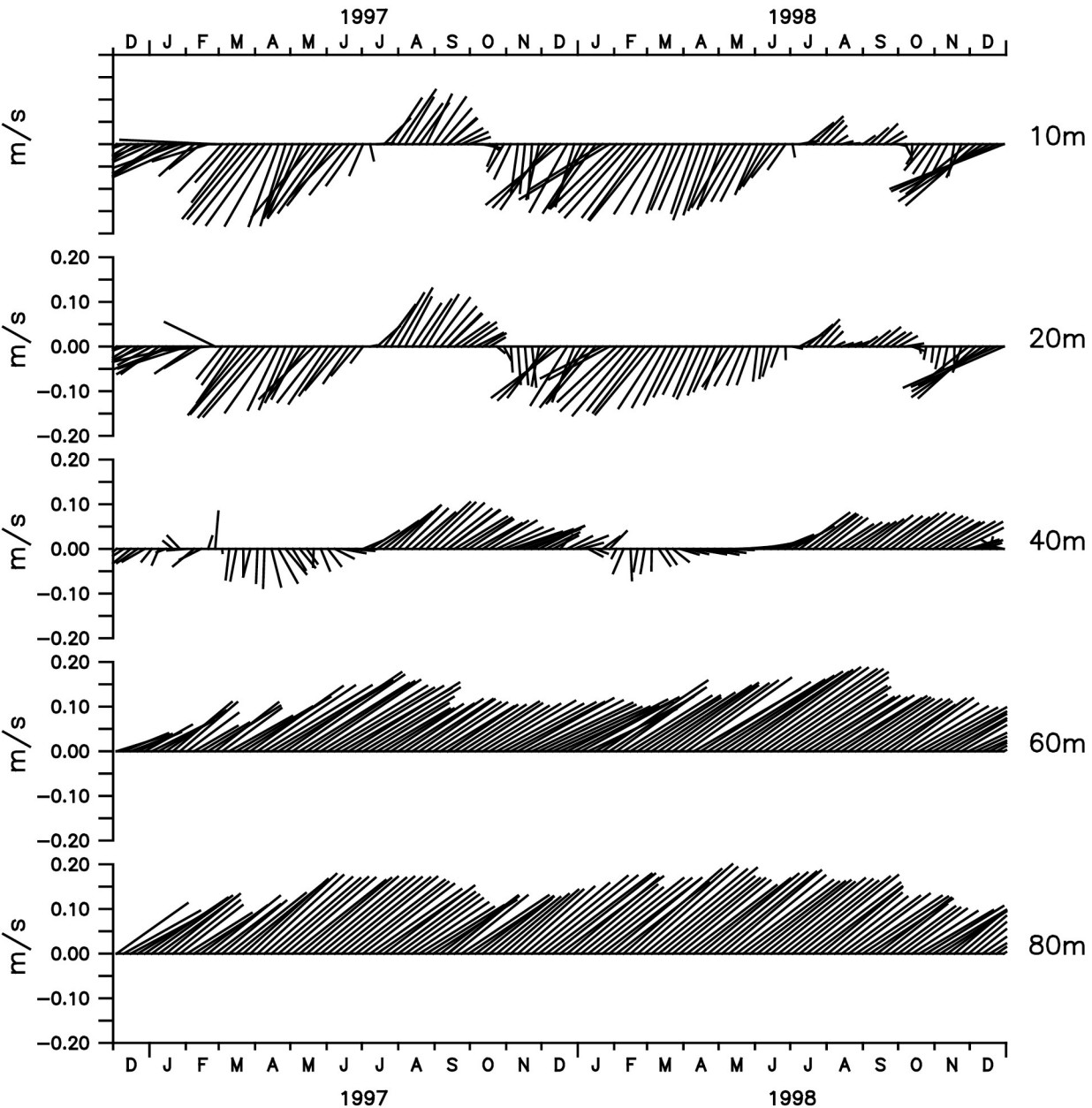

**Fig 7. Velocity stick vectors at different depts at the site of the *Johns et al. (2003)* [3] mooring, indicated in Fig 6.**

## 4.3 Volume and freshwater budgets

The diagnostics of the lateral exchanges start with the mean vertical structure of the meridional component of the velocity and salinity across the Hormuz section, in the Summer and Winter seasons (Fig 8). In spite of the seasonal difference, with a weakened northward flow during the winter (Fig 8), the two-layers configuration is present all year long, with the upper and lower layers separated by the $\sigma_2$ = 32.60 isopycnic surface. On the other hand, that specific surface is shallower in the western and deeper in the eastern side of the section. This supports the assumption that the baroclinic circulation can be divided in two components: one vertical overturning circulation, driven by the evaporation over the Gulf, and a local horizontal

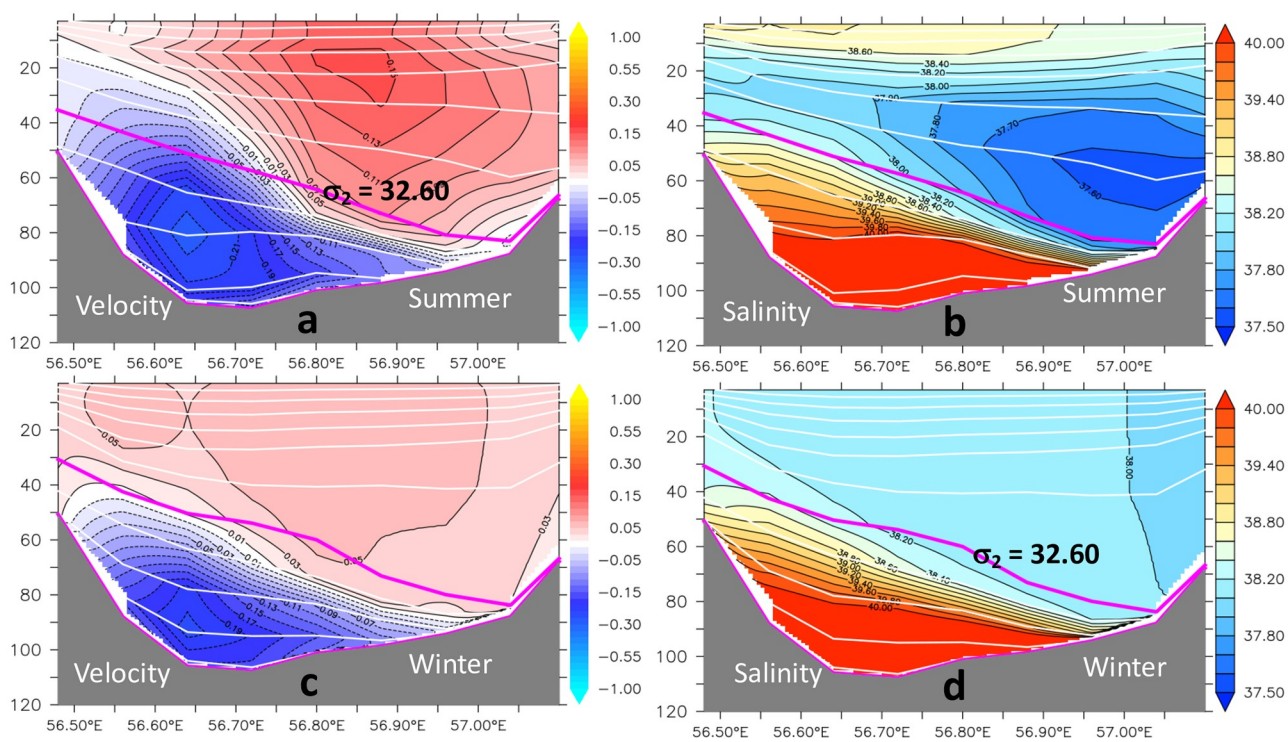

**Fig 8. Summer and winter distributions of the meridional velocity and salinity at the Hormuz section (26˚N).**

recirculation, similar to a gyre circulation, that could be, or not, directly related to the excess evaporation over the Gulf, as suggested by [3].

In this more complex baroclinic circulation pattern, the fresher waters from the Gulf of Oman cross the Hormuz section and enter the Persian Gulf as an upper-layer northward current, concentrated more to the eastern side of the section. The southward flow of saltier waters in the bottom layer is concentrated in the deeper part of the section, with the maximum values near the western boundary. Supperimposed on this overturning circulation, there is local horizontal recirculation, as indicated in Fig 6.

The first term in the right-hand side of Eq 1 is the total volume of water ($VT_H$) transported across a vertical section representing the Strait of Hormuz. The second term, *EMP*, is the net evaporation (evaporation minus precipitation). The last two terms are the contribution from river inflow and a residual term. In the Gulf, rivers represent only a small fraction in the water budget, being at the most, one order of magnitude smaller than *EMP* [3, 26, 28]. As a matter of fact, the river discharge is a moving target. During the thirty years of the simulations (1980—2015), many dams have been constructed in Turkey, Iraq and Iran, which have significantly decreased the freshwater discharge into the Gulf. These data are not freely available but the decreased riverine inflow should be adressed in future work. Another important subject that should be further investigated is the anthropogenic impacts on the E-P budget. There are suggestions (e.g.: [14–17]) that water desalination across the Gulf (removal of millions of cubic meters per day of freshwater and discharge of high salinity brines) may have a measurable impact on the salinity of the Gulf.

Assuming no change in the total volume with time, the rest term can me estimated as the difference between $VT_H$ and *EMP*. GLBa0.08 was forced with NCEP reanalysis products, which include monthly time series of precipitation. Evaporation minus precipitation is a

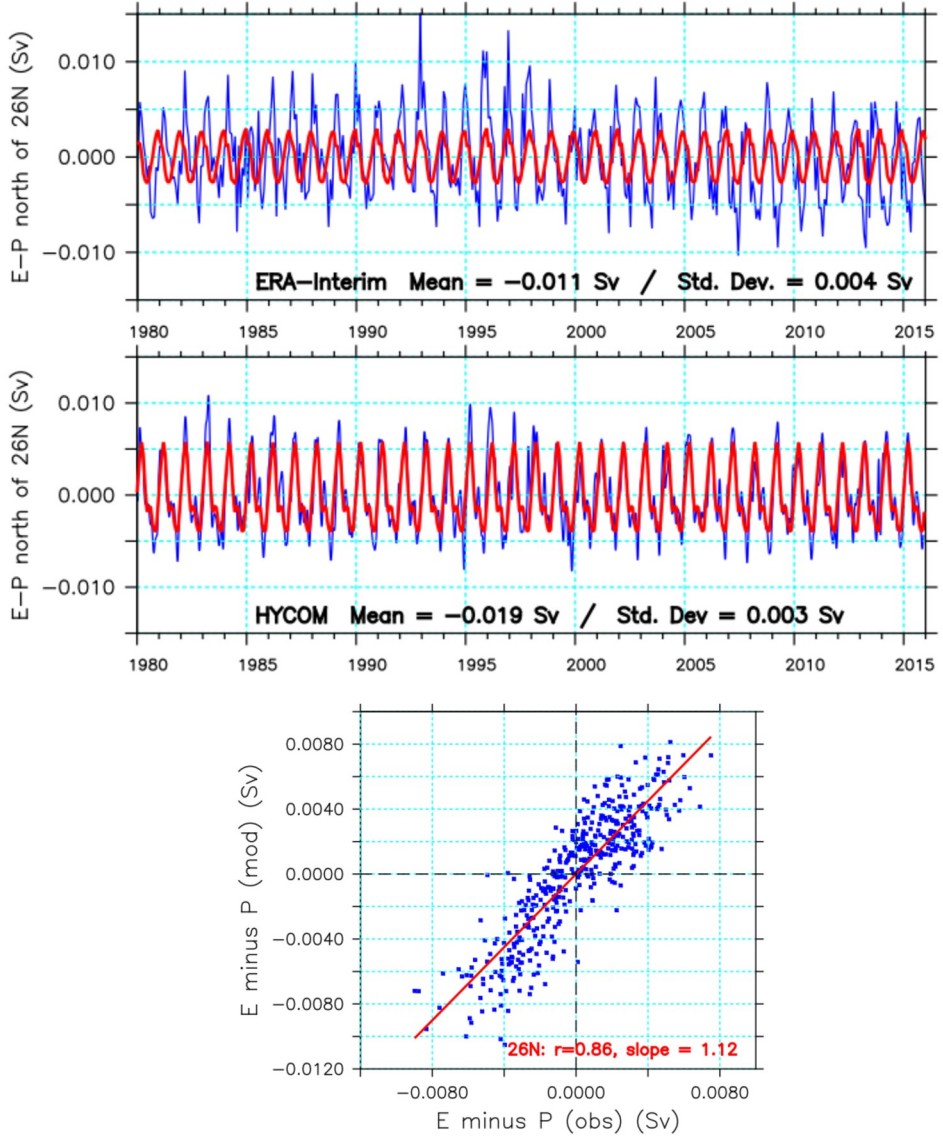

**Fig 9. Time series of *EMP* from ERA-Interin (top) and GLBa0.08 (middle), with the 6-day sampled values shown in blue and the seasonal cycles in red.** The bottom panel shows a scatter-plot of the two time-series.

model output and, in Eq 1, *EMP* is obtained by integrating over the entire region to the north of 26˚N, considering only water points.

To evaluate the accuracy of the *EMP* calculated by the model, a comparison was made with the net evaporation from the ERA-Interim reanalysis (Fig 9. Both ERA-Interim and GLBa0.08 products are marked by significant seasonal signals, with the somewhat different mean values of -0.011 *Sv* and -0.019 *Sv*, respectively. The standard deviations, however, are comparable: 0.004 *Sv* and 0.003 *Sv*, for ERA-Interim and GLBa0.08 respectively. The negative signal indicates loss of water due to excess evaporation, in both products. Despite the different mean values, however, the two-time series have variability with significantly good correlation (Fig 9), with $r^2 = 0.76$ and slope = 1.13. One could argue that the NCEP forcing and the ERA-Interim are both reanalysis products, having used common satellite and other ancillary data. However, it is still remarkable that the model's calculations do not drift away considerably from the

observations. The model's SST was relaxed to observations, with a 30-days e-folding time. The surface salinity was allowed to evolve freely, except in regions of river outflow, where the river discharge is input as precipitation, following approach described by [38]. In HYCOM, river inflow is treated like precipitation, parameterized as a virtual salt flux.

The time series of the total volume of water transported across Hormuz, $VT_H$, has also a marked seasonal variability, with mean value of 0.027 $Sv$ and higher standard deviation, 0.022 $Sv$, as compared with $EMP$. Considering these mean values ($VT_H = 0.027Sv$ and $EMP = -0.019Sv$), the mean error term ($RV$) should be equal to -0.008 $Sv$ for no volume trend in the Gulf. This is quite reasonable, since a considerable part of the salt transport across the lateral boundary (Hormuz) are due to mixing and smaller scale processes not resolved in the 1/12-degree horizontal resolution.

The calculation of $VT_H$, $M_{ov}$ and $M_a z$ were made after interpolating the layer velocity to the Cartesian vertical coordinate $z$. The total volume of water exported in the lower layer across the Hormuz section was calculated in the isopycnic coordinate space, with no interpolation. The mean value obtained was of 0.26±0.05$Sv$, which is similar to the 0.2$Sv$ value reported in the literature [3, 4].

Combining Eqs (1) and (2) and using the definitions from Eqs (3) to (8), one can obtain the total freshwater tendency in the Gulf:

$$M_t = EMP + M_{ov} + M_{az} + Rest, \tag{10}$$

where *Rest* represents all terms related to sources, sinks, mixing and truncation errors. A more realistic diagnostic of any tendency in the Gulf would require a more accurate knowledge of these terms. However, here only the contributions from the three first terms in the right-hand side of Eq 10 ($EMP$, $M_{ov}$ and $M_{az}$) are investigated in more details. Assuming no freshwater tendency ($M_t = 0$), Eq 10 can be rewritten as:

$$M_{ov} + M_{az} + Rest = -EMP. \tag{11}$$

Output from GLBa0.08, from 28-dec-1979 to 07-dec-2015 with a 6-days interval, where used to estimate $M_{ov}$ and $M_{az}$, using the Eqs 4 and 5. The resulting time series, $VT_H$, $M_{ov}$, $M_{az}$ and $EMP$ are plotted in Fig 10. $M_{ov}$ and $M_{az}$ have similar variability and mean values of 7.2±2.1 × 10$^{-3}$ $Sv$ and 5.0±1.7 × 10$^{-3}$ $Sv$ respectively. Their sum, the total baroclinic transport of freshwater across 26°N, is always positive and has mean value of 12.2 × 10$^{-3}$ $Sv$. The variability is marked by a strong seasonal signal, with some higher-frequency variability and significant year-by-year differences.

Inserting the mean values for $M_{ov}$, $M_{az}$ and $EMP$ in Eq 11, and assuming no salinity trend, one can obtain the mean residual term $Rest = 8 \times 10^{-3}$ $Sv$, which is equal to the residual term ($R_V$) in Eq 1, for no volume trend. Again, it is reasonable to assume that experiments with higher horizontal resolution and a better estimate of small scale mixing this residual term will be smaller and more accurate. This is very important because any imbalance could lead to salinity trends in the Gulf.

The correlations of the two components of the baroclinic freshwater transport, $M_{ov}$ and $M_{az}$, as well as of their sum ($M_{ov} + M_{az}$) with $EMP$ were computed for different time-lags. It was found that both components are reasonably well correlated, with the better results obtained for $EMP$ leading $M_{ov} + M_{az}$ by one month. As shown in the scatter-plots of Fig 11, the one-month lag values of $r^2$ with $EMP$ were 0.51, 0.55 and 0.59, respectively for $M_{ov}$, $M_{az}$ and $M_{ov} + M_{az}$ respectively. Here one must bear in mind that this relatively small correlation is likely to be due to the large residual term. In any case, the results indicate that evaporation over the Gulf drives salt export/freshwater import by both the overturning and horizontal

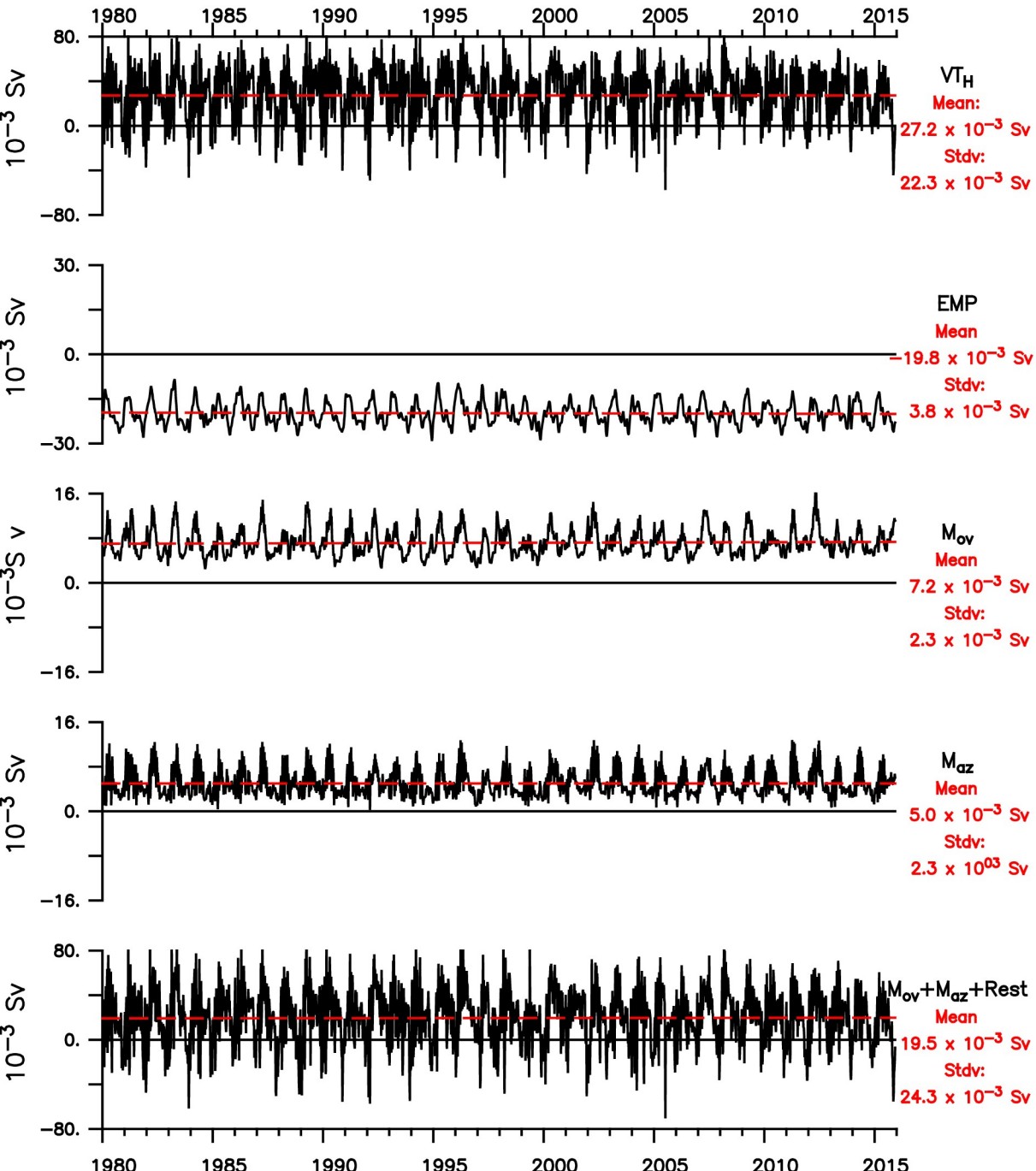

**Fig 10. From top to bottom: Time series of the total volume of water transported across the Hormuz section ($VT_H$); Evaporation minus precipitation over the Gulf ($EMP$), the overturning and horizontal components of the freshwater transport by the baroclinic circulation across Hormuz ($M_{ov}$ and $M_{az}$) and the sum of $M_{ov}$, $M_{az}$ and the residual term $R$.**

components of the internal baroclinic circulation. Improved investigations, using higher resolution models and more realistic forcing would yield more accurate results.

The time-series of the total freshwater exchange by the baroclinic circulation across the Strait of Hormuz ($M_{ov} + M_{az}$) is dominated by the seasonal cycle. To further investigate this

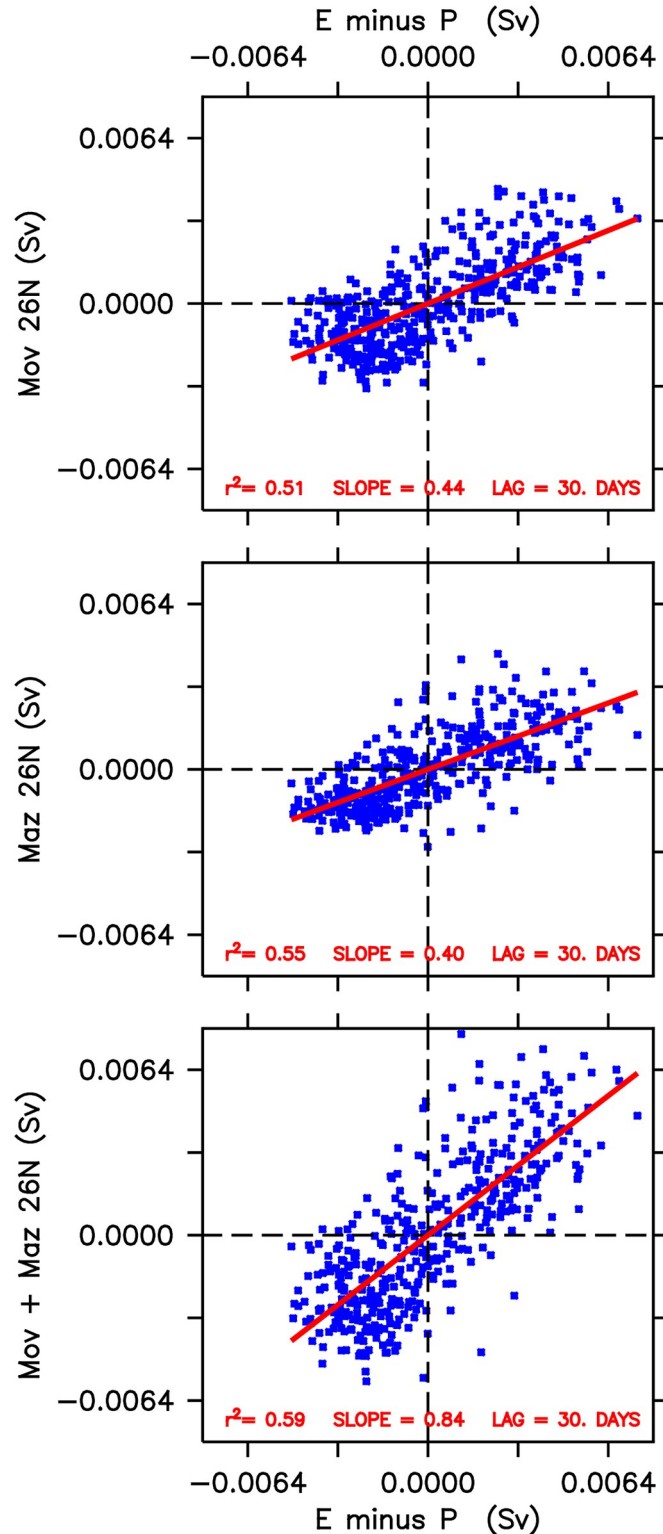

**Fig 11. Scatterplots btween *EMP* and the horizontal and overturing components of the baroclinic transport across Hormuz, at a one-month time lag.**

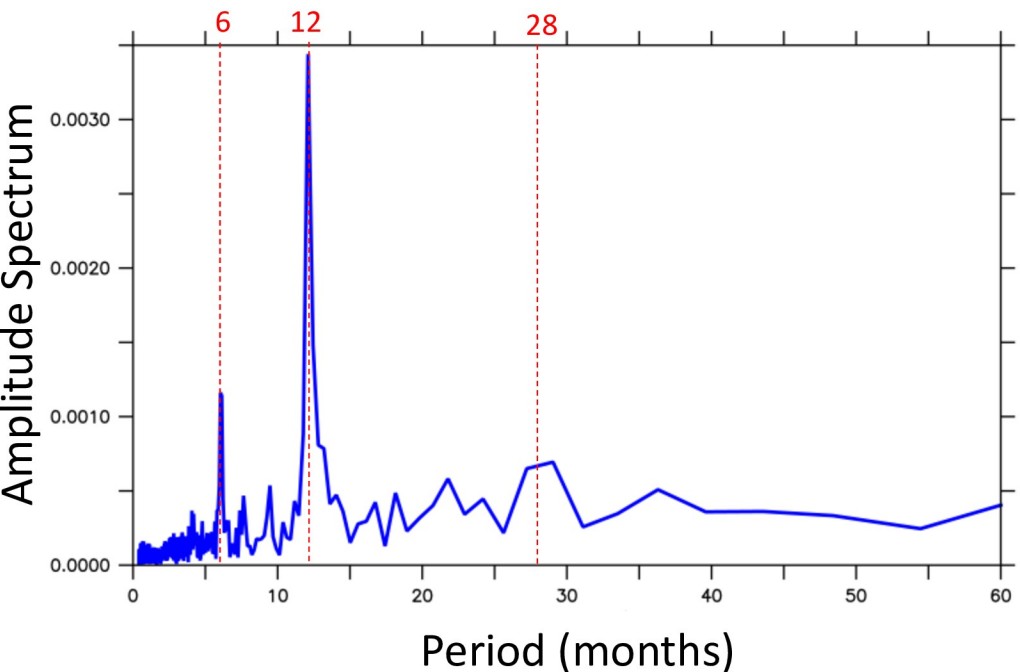

**Fig 12. Amplitude spectrum of the time-series of the total freshwater exchange by the baroclinic circulation across the Strait of Hormuz ($M_{ov} + M_{az}$).**

and to have a better assessment of its variability, the amplitude spectrum of the time series was computed using a Fast Fourier Transform (FFT) routine. The result, plotted in Fig 12, confirms that most of the energy is on the 12-months period variability. A second peak appears at 6 months, revealing a significant variability in the semi-annual period. This is certainly associated with the variability of the Shamal winds over the Gulf. A third peak, much smaller in amplitude and broader in the period domain, is seen in the range of 26 to 30 months. Although no significant correlation was found, one could argue that this third peak is a response to the ENSO signal.

## 5 Summary and conclusions

In spite of not being designed specifically to study a small semi-enclosed marginal sea, the general circulation pattern in the Persian Gulf produced by the 1/12-degree resolution experiment with a global implementation of HYCOM shows reasonably good agreement with previous modeling and observational work in that region. Based on that, the model results in the period from 1980 to 2015 were used to investigate the volume and freshwater budget over the Gulf and the lateral exchanges of freshwater with the Gulf of Oman, through the Strait of Hormuz.

The results show that the volume of the Persian Gulf Water exported in the lower layers into the Gulf of Oman, through the Straits of Hormuz, is of approximately 0.26 *Sv*, in good agreement with values reported by previous observational and numerical modeling studies [3, 4]. In the region were a current meter was deployed and maintained from 1/Dec/1996 to 1/Jan/1999 [3], the model presents a highly steady outward flow, in a very good agreement with the observations. In the upper layers, though, differently from the observations, the model yields a more variable flow pattern, and a predominant southwestward direction. However, during the JAS period, the surface circulation presents a nortweastward flow as part of a recirculation pattern that closes within the Gulf, suggesting that the surface waters do not reach the

eastern regions of the Strait of Hormuz, in the Gulf of Oman. This somehow answers the question whether the upper-layer recirculation observed by Johns *et al.* (2003) [3] would extend beyond the Musandam peninsula. In the lower layer, however, the model shows clearly a concentrated outlow near the Arabian Peninsula coastline and a somewhat weaker inflow along the Iranian coast, indicating the the baroclinic transport across the Strait of Hormuz has overturning and azimuthal components.

To evaluate the tri-dimensional structure of the freshwater transport across the Strait of Hormuz, the exchange across 26˚N was decomposed in an overturning and a horizontal component. It was found that both components are equally correlated with E-P over tht Gulf, at a one-month time lag. The highest correlation occurs for the total freshwater transport, the sum of the overturning and horizontal compoents, with the E-P over the Gulf leading. This suggests that the exchages by both overturning and horizontal circulation are driven by the dynamics and thermodynamics inside the Gulf.

The amplitude spectrum of the freshawater transport time series shows that most of the energy is in the seasonal cycle. A second peak appears at 6 months, revealing a significant variability in the semi-annual period, likely associated with the variability of the Shamal winds over the Gulf. The fast Fourier transform (fft) analysis also suggests some variability in the range of 26 to 30 months, what could be associated with the ENSO phenomena. In spite of the trends found in some of the forcing products, no significant trends were obtained in the model's results.

As stated previously, this paper is part of a continuing effort, aiming at the study of the pathways and spreading of waters from the Gulf into the Indian Ocean. Presently, the output of GLBa0.08 and ARGO data are being analyzed and the results being used in a second manuscript intended to be submitted for publication in the near future. Also, in the continuation, Lagrangian techniques togheter with higher-resolution nestings of the hydrodynamic model will be used to better understand the impacts of the smaller-scale processes.

## Acknowledgments

This paper is Lamont–Doherty Earth Observatory contribution number 8403.

## Author Contributions

**Conceptualization:** Edmo J. D. Campos, Arnold L. Gordon, Björn Kjerfve.

**Data curation:** Edmo J. D. Campos, Filipe Vieira.

**Formal analysis:** Edmo J. D. Campos, Arnold L. Gordon, Björn Kjerfve, Georgenes Cavalcante.

**Funding acquisition:** Edmo J. D. Campos, Georgenes Cavalcante.

**Investigation:** Edmo J. D. Campos, Arnold L. Gordon, Björn Kjerfve, Filipe Vieira, Georgenes Cavalcante.

**Methodology:** Edmo J. D. Campos.

**Project administration:** Edmo J. D. Campos.

**Validation:** Filipe Vieira.

**Visualization:** Edmo J. D. Campos, Filipe Vieira, Georgenes Cavalcante.

**Writing – original draft:** Edmo J. D. Campos.

**Writing – review & editing:** Edmo J. D. Campos, Arnold L. Gordon, Björn Kjerfve, Filipe Vieira, Georgenes Cavalcante.

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
