## [Decision Letter · Decision Letter 0]

24 Apr 2020

PONE-D-20-06928

Freshwater budget in the Persian (Arabian) Gulf and exchanges at the Strait of Hormuz

PLOS ONE

Dear Dr. Campos,

Thank you for submitting your manuscript to PLOS ONE. After careful consideration, we feel that it has merit but does not fully meet PLOS ONE’s publication criteria as it currently stands. Therefore, we invite you to submit a revised version of the manuscript that entirely addresses all the points raised by both reviewers during the review process.

We would appreciate receiving your revised manuscript by Jun 08 2020 11:59PM. To enhance the reproducibility of your results, we recommend that if applicable you deposit your laboratory protocols in protocols.io, where a protocol can be assigned its own identifier (DOI) such that it can be cited independently in the future. For instructions see: http://journals.plos.org/plosone/s/submission-guidelines#loc-laboratory-protocols

We look forward to receiving your revised manuscript.

Kind regards,

João Miguel Dias, Ph.D.

Academic Editor

PLOS ONE

Reviewers' comments:

Reviewer's Responses to Questions

**Comments to the Author**

1. Is the manuscript technically sound, and do the data support the conclusions?

Reviewer #1: Yes

Reviewer #2: Yes

2. Has the statistical analysis been performed appropriately and rigorously? 

Reviewer #1: Yes

Reviewer #2: Yes

3. Have the authors made all data underlying the findings in their manuscript fully available?

Reviewer #1: Yes

Reviewer #2: Yes

4. Is the manuscript presented in an intelligible fashion and written in standard English?

Reviewer #1: Yes

Reviewer #2: Yes

5. Review Comments to the Author

Reviewer #1: A nice research work investigating freshwater budget in the Persian Gulf and exchanges at the Strait of Hormuz using HYCOM model. The subject is interesting and within the aims and scope of PLOS ONE. Almost parts of the manuscript have been organized well. In addition, since the Persian Gulf is a hot spot in terms of climate change, more investigations are needed to justify the global warming effects on its hydrodynamics as well as chemical and biological components of the gulf. Therefore, i suggest acceptance after few minor revisions as described below:

1- What do you mean by “Sυ” in ABSTRACT? Please define it.

2- Large numbers of acronyms introduced in the manuscript makes difficulties for the readers to clearly read and understand it. I suggest the authors to remove some abbreviations such as RSW, UAE, and ... .

3- Line 42: Please give full name for SST.

4- The share of Persian Gulf and Red Sea in Indian Ocean water is so little that I think they do not change Indian Ocean water masses as you mentioned in the manuscript. They may affect some small parts of Indian Ocean in northwestern parts. If you do not believe that, please introduce some references to support your conclusion.

5- Section 5: I think it would be better if the authors could summarize this section and move some discussion made in the previous section.

Reviewer #2: The paper title “Freshwater budget in the Persian (Arabian) Gulf and exchanges at the Strait of Hormuz” uses a HYCOM model (which is generally used on larger water bodies) to explore the dynamics of the Persian Gulf and it’s inflow/outflow with the Gulf of Oman. The results are not earth shattering by any means – they largely align with results previously published elsewhere – but the compatibility of results does validate the use of the HYCOM approach in this shallow, enclosed sea. The authors do develop some novel findings related to the horizontal component of the exchange at the Strait of Hormuz. Detailed corrections are provided below, and my main comments are these (which are minor issues): 1) remove the term freshwater throughout and replace with normohaline or oceanic; freshwater implies hyposalinity; 2) I think you could do a better job of promoting what novel findings have come out of this research (particularly in the abstract) as many of the findings simply echo already well established patterns; 3) when referring to specific authors in text, add the authors name in the sentence, not just the reference numeral. OVERALL ASSESSMENT: A good quality paper that is suitable for publication after minor edits.

L4. OceanS

10. replace what with which.

10: greater depths than XXX m

18: better understand

22: Would suggest replacing reference 13 by adding these two more recent papers which discuss this issue further:

Lattemann S, Höpner T (2008) Impacts of seawater desalination plants on the marine environment of the Gulf. In Protecting the Gulf’s Marine Ecosystems from Pollution, Abuzinada A, et al., Editors. Birkhäuser Basel. 191-205. 10.1007/978-3-7643-7947-6_10

AND

Sale PF, Feary D, Burt JA, Bauman A, Cavalcante G, Drouillard K, Kjerfve B, Marquis E, Trick C, Usseglio P, van Lavieren H (2011) The growing need for sustainable ecological management of marine communities of the Persian Gulf. Ambio. 40(1):4-17. doi:10.1007/s13280-010-0092-6

23: Sea not Sean

24: Reference that these changes are common in RS?

31: add the name of the author, not just the numeral, as you are referring to a specific author in the text

33: ditto

35: ditto

42: ditto

45: Presumably these changes in SST were suggested to be due to climate change? If so, add that.

46-48: Above you mention that PGW is an important contributor to IO circulation, but nowhere do you say what the potential consequences may be if there are significant changes in PGW input. It would be useful to add a sentence to that effect here.

56-59: If it’s not covered here, I would suggest removing this. You can mention it as ongoing work in the Discussion.

68: salinity, not saline

71: smaller scales of length and time. Not sure what this means. Do you mean space and time?

72: explain what the shamal weather phenomenon is for readers not familiar.

74-94: Summer shamal winds are also important in driving the mid-Gulf eddies and transporting water to the southern basin (see Cavalcante). They also have ecological importance in transporting larvae (see Cavalcante) as well as evaporative cooling of the extremely hot summer sea temperatures (important for all marine fauna) (see Paparella).

Cavalcante GH, Feary DA, Burt JA (2016) The influence of extreme winds on coastal oceanography and its implications for coral population connectivity in the southern Arabian Gulf. Marine Pollution Bulletin. 105(2):489-497. doi:10.1016/j.marpolbul.2015.10.031

Paparella F, Xu C, Vaughan GO, Burt JA (2019) Coral Bleaching in the Persian/Arabian Gulf Is Modulated by Summer Winds. Frontiers in Marine Science. 6(205):1-15. doi:10.3389/fmars.2019.00205

107: I would suggest moving the precipitation data after the evaporation data so that things are given in the same order as the previous sentence.

107-115: Some units are in cm yr-1 and others in m3 s-1. It would be useful to get a relative understanding of approx.. how many cm yr-1 these would equate to (for comparative purposes), by calculating the addition of these inputs across the whole area of the Gulf by adding the following at the end of 113: “… , together equating to an approximately XXX cm yr-1 across the Gulf”.

119: replace in the bottom one with: along the bottom.

119: current _ meter

133-4: replace do not allow to infer with: do not allow inference of

138: replace is with: has been

145-6: combine these sentences.

151: add reference numeral

180: remove capital on Even

184: add ‘the’ before output

209: Here and elsewhere in the document you use the term ‘freshwater’ when what you mean is oceanic/normohaline water. Freshwater implies input from precipitation and river runoff to most. This needs to be fixed throughout (particularly in the title and several places in the abstract) as it is misleading. [side note: I did a search, and there are 43 instances where the term freshwater is used in the document; all need to be edited unless specifically referring to fresh (non-brackish) water].

218: add author name

226: ditto

33: ditto. AND HEREAFTER I will stop asking you to add the authors name when you are referring to a specific author in a sentence; please correct throughout document (i.e. not when it’s at the end of a sentence, but rather when you are, in text, referring to a specific author.

275: Hormuz

Fig 3. At least in the figure I am seeing, the diagram is so compressed it’s almost impossible to see the current speed/direction arrows other than in the eddy.

Fig 3 legend: What is the mean depth of the top 7 layers?

278: This finding is in keeping with Cavalcante and with Thoppil and Hogan

Cavalcante GH, Feary DA, Burt JA (2016) The influence of extreme winds on coastal oceanography and its implications for coral population connectivity in the southern Arabian Gulf. Marine Pollution Bulletin. 105(2):489-497. doi:10.1016/j.marpolbul.2015.10.031

Thoppil P, Hogan P (2010) A modeling study of circulation and eddies in the Persian Gulf. Journal of Physical Oceanography. 40:2122-2134.

Fig 4 legend: what is the mean depth of the layers?

Fig 5 legend: I assume the solid blue lines represent two standard deviations? If so, say so.

Fig 6. What does the red line represent? Add here, and in the later text, refer to it.

342: I’m assuming you have modeled based on a mean depth for 7 km grids in this area, correct? If so, what is the actual difference in bathymetric depth across some of your cells? This will help convey the magnitude of impact bathymetry may have, as that strait area does have some dramatic depth differences on those scales. This would help strengthen this argument.

343-4: to account for the higher frequency of the observations, could you reanalyze that data to give 6 day averages and then compare to see if this resolves the difference between the model and the observation? This would strengthen your argument.

364: MusandAm

364: seen, not seem

365: outward

393-398: There are also various authors out there suggesting that the amount of desalination is having a measurable impact on the salinity of the Gulf (removal of freshwater and discharge of high salinity brines, on the order of many millions of cubic meters daily across the Gulf). While I don’t agree with this hypothesis, personally, it might be something to consider mentioning.

483: spacing issue before numeral

485: where, not were

485 current _ meter

492: replace if with ‘of whether’

6. PLOS authors have the option to publish the peer review history of their article (what does this mean?). If published, this will include your full peer review and any attached files.

Reviewer #1: Yes: Roohollah Noori, PhD

Reviewer #2: No

---

## [Author Response · Author response to Decision Letter 0]

28 Apr 2020

REVIEWER #1

1) We inserted a definition for Sv (Sverdrup) in the abstract;

2) Following his suggestions, we reduced to a minimum the number of acronyms throughout the text, leaving only those deemed necessary and with the appropriate and clear definition;

3) As recommended, we inserted the definition for SST (sea surface temperature), at its first appearance in the text;

4) We modified the text, adding references and a few words about the potential importance of the Gulf waters in the Indian Ocean.

5) Section 5 was slightly reduced, following the reviewer’s recommendation.

REVIEWER #2

We sincerely acknowledge the excellent review, including detailed editing suggestions and comments throughout the manuscript, which we gladly accepted and contributed to the paper’s readability and quality. Below we itemize these changes, following the list in the reviewer’s comments. As for his major concerns, our response is the following.

1) The freshwater term. We apologize for not having made it clearer. This terminology is used because in the paper it is exactly hyposaline or water with zero salinity. The freshwater we refer to is the amount of water with no salt content that would correspond to the amount of salt transported in the opposite direction. Or, as explained by Drijfhout et al (2011): the equivalent freshwater budget obtained by dividing the equation for salt transport across a vertical section by a reference salinity (the quantity Mt in Eqs. 4 and 5, as explained on lines 265-266 of the original manuscript). To help the reader to better understand it, we included an explanation in the new version of the abstract.

2) We changed slightly the Abstract, to include some words making a better promotion of the most relevant findings. 

3) Thanks for pointing out this detail. We revised the complete manuscript and include the names when referring to specific authors.

Response to the reviewers #2 general comments:

Some of the points, listed below, were just editorial suggestions, typos or other minor comments. They were all accepted, with the appropriate change made in the new version.

Lines 4, 18, 22, 23, 24, 31, 33, 35, 42, 45, 46-48, 56-59, 68, 71, 72, 74-94, 107, 107-115, 119, 133-134, 138, 145-6, 151, 180, 184, 218, 226, 233, 275, 278, 364, 365, 483, 485, 492.

These are comments that required more elaborate responses.

Line 10: “what” was replaced with “which” and the verb was put in the plural (“make”) to agree with the subject: the warmer and more buoyant characteristics of the waters. 

Line 209: The freshwater issue, explained in Item (1) above

Fig. 3: We improved the figure and changed the caption, to include the layer mean depth.

Fig. 4: We included the mean depth in the caption and in the main text.

Fig 5: The blue lines were explained in the caption.

Fig 6: The red line explained in the caption.

Line 342: The reviewer is right. The model considers the mean depth in each 7km cell. However, in the vicinity of the point considered, the bathymetric gradient is not high. The major concern is with the bathymetry of a larger area, including the northernmost region of the Arabian Peninsula. The ETOPO5 bathymetry does not resolve it properly (we are currently running a version of the model, with a much higher horizontal resolution (1/48 of a degree) and improved bathymetry (GEBCO 15arcsec bathymetry). 

Lines 343-4: We apologize for not being clearer. We changed the text to better explain what we meant. Our model products are 6-day averages and the data collected by Johns et al (2003) have higher resolution. We do not have access to that data. In the future, we plan to analyze a higher frequency output of the higher-resolution model’s implementation.

Lines 393-398: We gladly accepted the suggestion and included some text with respect to the desalination activity. As a matter of fact, this is part of our ongoing investigation, in which we address the question.

---

## [Editor Report · Decision Letter 1]

29 Apr 2020

Freshwater budget in the Persian (Arabian) Gulf and exchanges at the Strait of Hormuz

PONE-D-20-06928R1

Dear Dr. Campos,

We are pleased to inform you that your manuscript has been judged scientifically suitable for publication and will be formally accepted for publication once it complies with all outstanding technical requirements.

With kind regards,

João Miguel Dias, Ph.D.

Academic Editor

PLOS ONE
---

## [Editor Report · Acceptance letter]

18 May 2020

PONE-D-20-06928R1 

Freshwater budget in the Persian (Arabian) Gulf and exchanges at the Strait of Hormuz 

Dear Dr. Campos:

I am pleased to inform you that your manuscript has been deemed suitable for publication in PLOS ONE. Congratulations! Your manuscript is now with our production department. 

With kind regards,

on behalf of

Dr. João Miguel Dias 

Academic Editor

PLOS ONE